Prevalence of Entamoeba species in captive primates in zoological gardens in the UK

Regan Carl S.
Yon Lisa
Hossain Maqsud
Elsheikha Hany M. hany.elsheikha@nottingham.ac.uk
School of Veterinary Medicine and Science, University of Nottingham , Loughborough, Leicestershire , UK
Braga Erika
Electronic publication date: 2014 Jul 29
Publication date: 2014
Volume: 2
Electronic Location ID: e492
Received 2014 Jun 12; Accepted 2014 Jun 30
Copyright: © 2014 Regan et al.
Copyright year: 2014
Copyright holder: Regan et al.
License: This is an open access article distributed under the terms of the Creative Commons Attribution License, which permits unrestricted use, distribution, reproduction and adaptation in any medium and for any purpose provided that it is properly attributed. For attribution, the original author(s), title, publication source (PeerJ) and either DOI or URL of the article must be cited.
License URL: https://creativecommons.org/licenses/by/4.0/

Keywords: Entamoeba, Homo sapiens, Zoonosis, Public health, Phylogenetics, Prevalence, Zoos, Nonhuman primates

Funding: Society for General Microbiology Elective Research Project Grant Undergraduate research project grant, University of Nottingham - School of Veterinary Medicine and Science Zebra Foundation student research grant Summer studentship was provided by the Society for General Microbiology, University of Nottingham and Zebra Foundation to Carl Regan. The funders had no role in study design, data collection and analysis, decision to publish, or preparation of the manuscript.

==============================
The aim of this study was to determine the prevalence of amoebic infection in non-human primates (NHPs) from six Zoological gardens in the United Kingdom. Initially, 126 faecal samples were collected from 37 individually identified NHPs at Twycross Zoo, UK, and were subjected to microscopic examination. A subsequent, nationwide experiment included 350 faecal samples from 89 individually identified NHPs and 73 unidentified NHPs from a number of UK captive wildlife facilities: Twycross Zoo (n = 60), Colchester Zoo (n = 3), Edinburgh Zoo (n = 6), Port Lympne Wild Animal Park (n = 58), Howletts Wild Animal Park (n = 31), and Cotswold Wildlife Park (n = 4). Samples were examined by PCR and sequencing using four specific primer sets designed to differentiate between the pathogenic E. histolytica, the non-pathogenic E. dispar, and non-pathogenic uninucleate cyst-producing Entamoeba species. In the first experiment, Entamoeba was detected in 30 primates (81.1%). Six (16.2%) primates were infected with E. histolytica species complex. The highest carriage of Entamoeba species was found in Old World Colobinae primates. In the nationwide experiment, molecular analysis of faecal samples revealed notable rates of Entamoeba infection (101 samples, 28.9%), including one sample infected with E. histolytica, 14 samples with E. dispar, and 86 samples with uninucleated-cyst producing Entamoeba species. Sequences of positive uninucleated-cyst producing Entamoeba samples from Twycross Zoo clustered with the E. polecki reference sequences ST4 reported in Homo sapiens, and are widely separated from other Entamoeba species. These findings suggest a low prevalence of the pathogenic Entamoeba infection, but notable prevalence of non-pathogenic E. polecki infection in NHPs in the UK.

Introduction

Entamoeba (family Entamoebidae) is a genus of diverse intestinal protists found in humans, nonhuman primates (NHPs) and other animals. It encompasses several species, including E. histolytica, E. dispar, E. moshkovskii, E. polecki, E. nutalli, E. chattoni, E. coli, E. hartmanni, E. ecuadoriensis and E. Bangladeshi. NHPs harbour a number of Entamoeba spp. of varied importance to human and domestic animal health. E. histolytica species complex (E. histolytica, E. dispar and E. moshkovskii) are morphologically indistinguishable, but have different virulence capabilities. E. histolytica is the most important zoonotic pathogen (Sargeaunt, Williams & Jones, 1982; Verweij et al., 2003; Ekanayake et al., 2006), and has been reported in NHPs, causing intra- and extra-intestinal disease (Solaymani-Mohammadi et al., 2006; Ulrich et al., 2010). Also, E. histolytica is known to be responsible for 50 million human cases of haemorrhagic colitis and extra-intestinal abcessation, and 100,000 deaths annually (World Health Organization, 1997). In contrast, E. dispar is able to colonize the intestine, but is noninvasive. E. moshkovskii is primarily free-living and the ability to cause disease in human is still unclear (Heredia, Fonseca & Lopez, 2012). Also, human diseases linked to the uninucleated cyst-producing E. chattoni have been attributed to contact with monkeys (Sargeaunt, Patrick & O’Keeffe, 1992). E. nuttalli and E. histolytica species complex have previously been confused or misidentified on routine examination due to their morphological similarity, but are now considered separate species with restricted host specificity (Tachibana et al., 2013).

Microscopic examination of faecal samples has been traditionally the primary method of Entamoeba detection; however, it does not allow the differentiation of the pathogenic E. histolytica from the non-pathogenic Entamoeba spp (Kebede et al., 2003; Verweij et al., 2003; Fotedar et al., 2007). Knowledge of Entamoeba epidemiology and evolution has considerably progressed in recent years, with improved isolation, identification, and genotyping methods (Levecke et al., 2010; Stensvold et al., 2011). These molecular methods have detected considerable diversity within the genus, and enabled the detection and distinction of species (including the so-called ribosomal lineages) that cannot be differentiated by traditional parasitological methods. Despite the continued importance of Entamoeba spp and the known susceptibility of NHPs to infection very little information is available on the prevalence of Entamoeba infection in NHP populations in the United Kingdom. Given its zoonotic potential and public health impact, including in a zoological setting the present study assessed and compared Entamoeba prevalence in captive primates in various zoological gardens throughout the United Kingdom using molecular methods of Entamoeba detection.

Materials & methods

Study areas and sampling design

A preliminary study was performed to establish the prevalence of Entamoeba infection in primates in a single zoological park in the United Kingdom, and to identify which families of primates required the most focus during the subsequent nationwide study. Hence, Twycross Zoo was chosen as the preliminary study site as it houses a wide variety of primate species and families, and amoebic infection had been identified historically and was suspected in their primates at the time of the study. Thirty-seven primates were available for inclusion within the study, including six species of primates from 23 enclosures (Table 1). To identify individual primates in group enclosures a feed item for each primate was impregnated with approximately 0.5 g of different coloured edible cake glitter (Rainbow Dust Colours Limited, Lockstock Hall Preston, England) and fed during the morning by the keeper. Each group individual was assigned and fed a different glitter colour from two days before sample collection until successful completion of sample collection; this was typically two to five days. All stool samples in each enclosure were collected separately on clean disposable paper plates during morning cleaning by keepers until three stools per primate were identified from the glitter colour allocated for each primate. Three samples per primate were collected to account for intermittently shed Entamoeba. A representative 3–5 g stool sample for each animal (identified by different glitter colours) was placed into a labeled clean 7 ml plastic bijoux, using a clean wooden swab stick, containing 10% formalin, and was stored at 5 ° C until further processing. Age, sex, species, treatment with amoebicidal medication and enclosure identity were recorded.

Table 1 The prevalence of Entamoeba spp. in non-human primates from Twycross Zoo.

Species	Family	E. histolytica complex	E. hartmanni	E. coli	
Black Howler Monkey (Alouatta caraya; n = 17)	Atelidae	11.8% (2)	35.3% (6)	70.6% (12)	
Brown Woolley Monkey (Lagothrix lagotricha; n = 6)	Atelidae	16.7% (1)	66.7% (4)	66.7% (4)	
Eastern Javan Langur (Trachypithecus auratus auratus; n = 9)	Colobinae	22.2% (2)	88.9% (8)	77.8% (7)	
Dusky Leaf Monkey (Trachypithecus obscures; n = 1)	Colobinae	100% (1)	100% (1)	0%	
Golden Lion Tamarin (Leontopithecus rosalia; n = 2)	Callitrichidae	0%	0%	0%	
Golden-headed Lion Tamarin (Leontopithecus chrysomelas; n = 2)	Callitrichidae	0%	0%	0%	
Total (n = 37)		16.2% (6)	51.4% (19)	62.2% (23)	

Subsequent to the preliminary study, a nationwide epidemiological study was conducted to identify the prevalence of Entamoeba infection, E. histolytica, E. dispar and uninucleate cyst-producing species, within Colobinae primates at six different zoos in the United Kingdom (Fig. 1). Primates from the Colobinae family (genus Semnopithecus, Trachypithecus or Presbytis) were selected as the sample population for the nationwide study based on two reasons. Firstly, the preliminary data from Twycross Zoo demonstrated the highest prevalence of amoebiasis in the Old World Colobinae monkeys. This is in agreement with results from other studies (Tachibana et al., 2001; Levecke et al., 2007). Secondly, primates from the Colobinae family have specialised sacculated stomachs, an adaption to their leaf-eating lifestyle, which provides favorable conditions for ingested Entamoeba cyst excystation, and trophozoite tissue invasion (Mätz-Rensing et al., 2004; Ulrich et al., 2010).

Figure 1 Map of The United Kingdom showing the sampling locations.

Six zoological gardens are indicated by red solid stars. The map was created by using the STEP MAP web tool. 1 Twycross Zoo, Atherstone, Midlands, CV9 3PX, England. 2 Port Lympne Wild Animal Park, Lympne, Hythe, Kent, CT21 4LR, England. 3 Howletts Wild Animal Park, Bridge, Canterbury, CT4 65AE, England. 4 Colchester Zoo, Stanway, Colchester, Essex, CO3 0SL, England. 5 Cotswold Wildlife Park, Bradwell Grove, Burford, Oxfordshire, OX18 4JP, England. 6 Edinburgh Zoo, Edinburgh, City of Edinburgh, E12 6TS, Scotland.

A total 350 samples were collected from 162 primates from six zoological parks within the United Kingdom (Table 2), between July 2010 and August 2011. This sample group included primarily primates from the Colobinae family, but also some New World monkeys. All zoological parks housed primates and non-primate species. Primates occupied indoor concrete enclosures with access to external grassed sections. Same species primates occupied mixed sex group enclosures. Some primates were housed alone for medical reasons, or due to social incompatibility with the rest of the group. It was possible to collect repeat samples from four primates sampled in the preliminary study; all other primates from Twycross Zoo were unavailable for sampling. The same stool collection technique was used as for the preliminary study with one modification to facilitate molecular examination of samples: stools were collected into 70% ethanol, not 10% formalin. The primate keepers at each facility administered the glitter and collected the samples. Two hundred and seventy-four stool samples could be associated with 89 individually identified primates; however, some stool samples collected could not be attributed to a specific primate from within a group enclosure. This was due to the limitations of deciphering different glitter colours when dealing with large number of primates, and hence glitter colours, in one enclosure. Hence, 76 stools from the remaining 73 primates had to be collated as samples from eleven groups of NHPs (Table 2). The entirety of each stool sample was examined grossly for the presence of blood as a possible indication of gastrointestinal illness and potential parasitism. Thirty two (82.1%) of primates had been treated with a vitamin D3 supplement and 10 day course of metronidazole (Flagyl) followed by 10 days of diloxinide furoate in the six months prior to sample collection.

Table 2 Non-human primates sampled in the nationwide study.

Study site	Species of primate	Number of primates sampled	
		Individually
identified	Unidentified primates
(no. of group)	
Twycross Zoo	Eastern Javan Langur (Trachypithecus auratus auratus)	9	4 (1)	
	Black Howler Monkey (Alouatta caraya)	15	11 (2)	
	Woolley Monkeys (Lagothrix lagotricha)	6	–	
	Dusky Leaf Monkey (Trachypithecus obscures)	1	–	
	Golden Lion Tamarin (Leontopithecus rosalia)	2	–	
	Golden-headed Lion Tamarin (Leontopithecus chrysomelas)	2	–	
	Francois Langur (Trachypithecus francoisi)	5	–	
	Dusky Leaf Monkey (Trachypithecus obscures)	5	–	
Port Lympne Wild Animal Park	Eastern Javan Langur (Trachypithecus auratus auratus)	9	39 (4)	
	Grizzled Leaf Monkey (Presbytis comata)	–	7 (1)	
	Banded Leaf Monkey (Presbytis femoralis)	–	3 (1)	
Howletts Wild Animal Park	Banded Leaf Monkey (Presbytis femoralis)	5	–	
	Dusky Leaf Monkey (Trachypithecus obscures)	14	–	
	Francois Langur (Trachypithecus francoisi)	2	–	
	Grizzled Leaf Monkey (Presbytis comata)	8	–	
	Eastern Javan Langur (Trachypithecus auratus auratus)	2	–	
Colchester Zoo	Silvery Langur (Trachypithecus cristatus cristatus)	3	–	
Cotswold Wildlife Park	Purple-faced Langur (Trachypithecus vetulus monticola)	1	3 (1)	
Edinburgh Zoo	Purple-faced Langur (Trachypithecus vetulus vetulus)	–	6 (1)	
Total		89	73 (11)	

The study was approved by The University of Nottingham (UK) School of Veterinary Medicine and Science (SVMS) Ethical Review Committee. The Committee reviews all research studies involving School personnel and is chaired by Professor David Haig. The committee passed this study as good to proceed, not requiring any further ethical review.

Parasite identification

All formalin preserved samples were analysed microscopically using a modified Ridley’s formol-ether concentration technique, which enhances microscopic sensitivity by producing ‘cleaner’ samples that are more efficient to examine. Following sedimentation, samples were then examined microscopically for the presence of Entamoeba species from the E. histolytica complex (E. histolytica, E. dispar and E. moshkovskii), E. coli and E. hartmanni. Data was analyzed using Minitab 15. Binary logistic regression was used to demonstrate statistical significance between prevalence of infection and primate demographics. All prevalence data is derived using the total number of primates as the denominator.

Molecular analyses

QIAamp DNA Stool Mini Kit (QIAgen, UK) was used according to the manufacturer’s instructions to extract parasite DNA directly from faeces. Technique modifications to improve the yield and purity of DNA extracts included increasing the lysis temperature to 95 ° C and adding an extra wash prior to sample elution with Buffer AE. Concentration and DNA purity in sample extracts was analyzed, using a Thermo Scientific NanoDrop™1000 Spectrophotometer, prior to PCR amplification. The strategy used for selection of PCR primers (Table 3) was based on the use of previously published diagnostic primers for the mononucleate Entamoeba species, E. histolytica and E. dispar (Ali, Zaki & Clark, 2005). Both species have been previously found in the faeces of NHPs. It is important to discriminate E. histolytica from other nonpathogenic amoebas because E. histolytica carries the risk of zoonosis (Rivera & Kanbara, 1999; Tachibana et al., 2000; Tachibana et al., 2001; Verweij et al., 2003; Rivera, Yason & Adao, 2010; Feng et al., 2011). Also, we used two species complex specific primers to amplify uninucleate cyst-producing species, but not tetra- or octonucleate cyst-producing species. All PCR products were subjected to DNA sequencing to identify the species/subtype of each amplicon including those amplified by the species diagnostic primers.

Table 3 Primer sets and PCR conditions used in the present study.

	Forward primer	Reverse primer	Amplification reaction	
Primer set 2*	Primer 5.1: (5″-AAG GAT AAC TCT TGT TAA TTG CAG-3″)	Primer 3.2: (5″-TGT CTA AAT TAC CCC AAT TTC C-3″)	30 cycles of 94 ° C, 57 ° C, and 72 ° C each for 30 s, followed by a final 2 min at 72 ° C	
Primer set 3*	Primer 5.2: (5″-GGA ATA GCT TTT TGA GAA GAA GG-3″)	Primer 3.2: (5″-TGT CTA AAT TAC CCC AAT TTC C-3″)	30 cycles of 94 ° C, 57 ° C, and 72 ° C each for 30 s, followed by a final 2 min at 72 ° C	
E. histolytica	
	RRH5: (5″-GCG CCT TTT TAT TCA ATA TAC TCC-3″)	RRH3: (5″-GGA TGA AGA TAT CTT CAC AGG G-3″)	30 cycles of 94 ° C, 59 ° C , and 72 ° C each for 30 s, followed by a final 2 min at 72 ° C	
E. dispar	
	RRD5: (5″-CAT GAG GCG CCT TTT TAT CA-3″)	RRD3: (5″-AGG GGA TGA TGA TAT TGA ACA CAC TC-3″)	30 cycles of 94 ° C, 59 ° C , and 72 ° C each for 30 s, followed by a final 2 min at 72 ° C	
Notes.

* Two primer sets were used to target uninucleated cyst-producing Entamoeba species (E Victory, pers. comm., 2010).

Separate PCRs were performed with each primer pair in a reaction mixture of 40 µ l consisting of 4 µ l of extracted DNA, 20 µ l of Biomix (Bioline, UK), 15 µ l of sterile distilled water, and 0.5 µ l of each forward and reverse primer. The amplification reactions were performed using a Bioer Xp Cycler as described in Table 4. PCR products were separated by electrophoresis in 1.2% agarose gels run at 100 V on a Thermo Scientific Easycast B1 or D2 electrophoresis gel tank with a Thermo Scientific EC 1000 XL Power Pac for approximately 60 min. A mix of 7 µ l of PCR product and 3.5 µl of loading buffer (New England Biolabs Ltd., UK) were applied to each well. A 1-kbp molecular size ladder (New England Biolabs) was added to each gel for product size estimation. Gels were stained with 0.1 µ g/ml ethidium bromide solution. Amplified DNA was visualized under UV light.

Table 4 Details of purified amplicons of Entamoeba species from which nucleotide sequences were obtained.

Primate species	Zoological park	Primers	Target	
Banded Leaf Monkey (Presbytis femoralis)	Howlett’s Wild Animal Park	RRH3, RRH5	E. histolytica	
Eastern Javan Langur (Trachypithecus auratus auratus)	Twycross Zoo	RRD3, RRD5	E. dispar	
Dusky Leaf Monkey (Trachypithecus obscures)	Twycross Zoo	RRD3, RRD5	E. dispar	
Dusky Leaf Monkey (Trachypithecus obscures)	Howletts Wild Animal Park	RRD3, RRD5	E. dispar	
Eastern Javan Langur (Trachypithecus auratus auratus)	Howletts Wild Animal Park	RRD3, RRD5	E. dispar	
Eastern Javan Langur (Trachypithecus auratus auratus)	Twycross Zoo	RRD3, RRD5	E. dispar	
Eastern Javan Langur (Trachypithecus auratus auratus)	Twycross Zoo	RRD3, RRD5	E. dispar	
Eastern Javan Langur (Trachypithecus auratus auratus)	Twycross Zoo	RRD3, RRD5	E. dispar	
Eastern Javan Langur (Trachypithecus auratus auratus)	Twycross Zoo	RRD3, RRD5	E. dispar	
Black Howler Monkey (Alouatta caraya)	Twycross Zoo	P3.2, P5.2	Uninucleates	
Black Howler Monkey (Alouatta caraya)	Twycross Zoo	P3.2, P5.2	Uninucleates	
Woolly Monkey (Lagothrix lagotricha)	Twycross Zoo	P3.2, P5.2	Uninucleates	
Golden-headed Lion Tamarin (Leontopithecus chrysomelas)	Twycross Zoo	P3.2, P5.2	Uninucleates	
Eastern Javan Langur (Trachypithecus auratus auratus)	Twycross Zoo	P3.2, P5.2	Uninucleates	
Eastern Javan Langur (Trachypithecus auratus auratus)	Twycross Zoo	P3.2, P5.2	Uninucleates	
Woolly Monkey (Lagothrix lagotricha)	Twycross Zoo	P3.2, P5.2	Uninucleates	

Molecular phylogenetics

One positive amplicon per primate (a total of 16 amplicons) was selected for sequencing, based on visualization of PCR products (Table 4). Amplicons were purified, using a QIAquick PCR Purification Kit (QIAgen, UK), according to the manufacturer’s instructions and then subjected to sequencing on the Illumina platform by Source BioScience (Nottingham, UK) using the primers from the PCR. Nucleotide sequences were determined at least once on each DNA strand. Three representative Entamoeba nucleotide sequences obtained in this study were deposited in GenBank under accession numbers KJ149294, KJ149295, KJ149296.

Raw sequencing chromatograms were evaluated with Geneious (version 5.4) software. Newly obtained Entamoeba sequences were compared with similar sequences available at the GenBank database by using the Bl2Seq algorithm as implemented in BLASTn (Altschul et al., 1990). Multiple alignments of all nucleotide sequences were obtained by using the MUSCLE program (Edgar, 2004). The resulting alignments were adjusted manually when necessary using CLUSTALX (Larkin et al., 2007). The unmatched ends were deleted to obtain a homogeneous matrix of characters and thus increase the reliability of the tree obtained. Phylogenetic trees were inferred from the nucleotide sequence alignments by the maximum-likelihood (ML) method using the BIONJ algorithm (Gascuel, 1997) and distance method with HKY85 model (Hasegawa, Kishino & Yano, 1985) of nucleotide substitution implemented in PhyML-aLRT (Guindon & Gascuel, 2003). The reliability of the branching order was assessed by performing 1,000 bootstrap replicates.

Results

Entamoeba prevalence at Twycross Zoo

One hundred and twenty-six stool samples were collected from 37 individual primates. No primate demonstrated ill health at the time of sample collection and no samples contained grossly visible blood. Microscopic examination demonstrated Entamoeba shedding in 81.1% of 37 primates sampled (Table 1). Entamoeba coli was the most prevalent Entamoeba species shed (62.2%), with three of six primate species shedding this Entamoeba species. Shedding of species from the E. histolytica complex was identified in 16.2% of primates (6 primates). Co-infection with two or more Entamoeba species was identified in 14 primates. Old World Colobinae primates showed the highest prevalence of Entamoeba infection. Entamoeba infection was significantly associated with species of primate (P < 0.05) and administration of metronidazole (P < 0.05). More specifically, infection with E. coli was significantly associated with both parameters (both P < 0.05). Primates previously treated with metronidazole showed greater infection with E. coli (76.9%) compared to those untreated (25.0%). No significant associations were identified between primate demographic characteristics and infection with Entamoeba from the E. histolytica complex or E. hartmanni. Eggs from Trichuris species were identified in samples from two primates.

Entamoeba prevalence at multiple zoos

Entamoeba was present in 101 (28.9%) samples (Table 5), indicating a notable prevalence of Entamoeba infection at the national level. No more than one species of Entamoeba was identified per sample. Three Entamoeba species were detected by species-specific PCR and confirmed with sequencing and BLAST: E. histolytica, E. dispar and E. polecki. E. histolytica was detected in one sample (2.9%), E. dispar in 14 samples (4.0%) and uninucleated cyst-producing Entamoeba species in 86 (24.6%) samples. E. histolytica and E. dispar were identified in samples from Colobinae primates only, whilst uninucleated-cyst producing Entamoeba species were identified in samples primarily from New World monkeys, but also in primates from the Colobinae family. Entamoeba infection was only detected in primates from three zoological parks (Table 5): E. histolytica was only identified at one park, E. dispar in three parks, and uninucleated cyst-producing Entamoeba species in three parks. No primate was found to harbor mixed Entamoeba species. All primates sampled appeared clinically healthy at the time of sample collection.

Table 5 The prevalence of Entamoeba species by species of primate and zoological park.

Species of primate	E. histolytica	E. dispar	Uninucleates	
	% (number of primate)	
Old world monkey	
Banded Leaf Monkey (Presbytis femoralis; n = 20)	5.0 (1)	0	0	
Dusky Leaf Monkey (Trachypithecus obscures; n = 60)	0	3.3 (2)	0	
Eastern Javan Langur (Trachypithecus auratus auratus; n = 117)	0	10.3 (12)	29.1 (34)	
Francois Langur (Trachypithecus francoisi; n = 19)	0	0	0	
Grizzled Leaf Monkey (Presbytis comata; n = 27)	0	0	0	
Silvery Langur (Trachypithecus cristatus cristatus; n = 9)	0	0	0	
Purple-faced Langur (Trachypithecus vetulus vetulus; n = 11)	0	0	0	
Subtotal (n = 263)	0.76 (1)	5.3 (14)	12.9 (34)	
New world monkey	
Black Howler Monkey (Alouatta caraya; n = 52)	0	0	70.9 (40)	
Woolly Monkey (Lagothrix lagotricha; n = 23)	0	0	47.8 (11)	
Golden-headed Lion Tamarin (Leontopithecus chrysomelas; n = 6)	0	0	16.7 (1)	
Golden Lion Tamarin (Leontopithecus rosalia; n = 6)	0	0	0	
Subtotal (n = 87)	0	0	59.8 (52)	
Total (n = 350)	0.3 (1)	4.0 (14)	24.6 (86)	
Zoological park				
Colchester Zoo (n = 9)	0	0	0	
Cotswold Wildlife Park and Gardens (n = 8)	0	0	0	
Edinburgh Zoo (n = 3)	0	0	0	
Howletts Wild Animal Park (n = 90)	1.1 (1)	2.2 (2)	0	
Port Lympne Wild Animal Park (n = 72)	0	2.8 (2)	0	
Twycross Zoo (n = 168)	0	6.0 (10)	51.2 (86)	
Total (n = 350)	0.3 (1)	4.0 (14)	24.6 (86)	

To infer the phylogenetic relationship of the isolates detected in the present study, E. histolytica, E. dispar and E. polecki, with previously characterized isolates, we used maximum likelihood method. PCR amplicons from sixteen samples were purified and submitted for sequencing (Table 4). The sequence from the sample that produced an amplicon with the E. histolytica-specific primers was identical to the corresponding region of the GenBank sequence for E. histolytica from monkey (AB197936) from a cynomolgus monkey. Likewise, sequences obtained from eight samples that produced amplicons with the E. dispar-specific primers were identical to the corresponding region of the GenBank sequence (AB282661) for E. dispar from a rhesus monkey. Seven sequences were obtained from uninucleate amplicons from Twycross Zoo and shared high sequence homology E. polecki. Two representative sequences were used to build a phylogenetic tree. As seen in Fig. 2, E. polecki sequences obtained in the present study clustered with and formed a monophyletic group with E. polecki subtype 4 isolates reported in Homo sapiens from Asia, Africa and Europe.

Figure 2 Phylogenetic tree based on partial 18SrDNA sequences, showing the relationships among Entamoeba species.

Phylogenetic analysis used two different approaches, distance-based analysis and maximum-likelihood (ML), produced trees with identical topologies of which only ML tree is presented. GenBank accession numbers and host species are given in parentheses after the taxon name. Sequences in bold face were obtained during this study. Numbers above branches are bootstrap values (%) from 1,000 replicates. Nodes of the tree with bootstrap values of ≥95% are indicated by black closed circles. The node is not labeled where bootstrap support values is <50. Bar = estimated number of substitutions per site.

Discussion

Nonhuman primates harbour a number of Entamoeba spp of varied importance to human and domestic animal health. The prevalence and genetic identity of Entamoeba species was investigated in primate collections at six major NHP zoos in the United Kingdom. Results indicated a low prevalence of the pathogenic E. histolytica in the examined primates. This is important to the primate population and also to the many thousands of human visitors of these zoos each year. Higher prevalence of non-pathogenic Entamoeba species was however identified in the primates sampled. Previous studies utilizing molecular methods to identify carriage of Entamoeba species demonstrated a similar prevalence data to that seen in the current study. Low carriage of E. histolytica and higher carriage of other Entamoeba species in NHP populations has been demonstrated in both captive (Tachibana et al., 2000; Tachibana et al., 2001; Takano et al., 2005; Rivera, Yason & Adao, 2010) and free-living NHP species (Rivera & Kanbara, 1999). However, the NHP populations examined in these studies were based outside of Europe, with all of the captive populations investigated in these studies existing in research facilities in Asia. Levecke et al. (2010) reported that 36% of faecal samples collected from various primate species in zoological parks in Belgium and The Netherlands contained E. histolytica, and identified Entamoeba species as the most prevalent gastrointestinal parasite within the sampled population.

The lack of sex or age predisposition to infection with Entamoeba species in our study is in agreement with other studies (Lilly, Mehlman & Doran, 2002; Jones-Engel et al., 2004; Gillespie, Greiner & Chapman, 2005; Muehlenbein, 2005; Ekanayake et al., 2006; Teichroeb et al., 2009). In the present study, the highest prevalence of Entamoeba infection was detected in Old World monkeys; this finding is in agreement with reports from other studies in Japan (Tachibana et al., 2001) and Belgium (Levecke et al., 2007). Primates from the Colobinae family have specialised sacculated stomachs, an adaption to their leaf-eating lifestyle, which provides favorable conditions for ingested Entamoeba cyst excystation, and trophozoite tissue invasion (Mätz-Rensing et al., 2004; Ulrich et al., 2010). The higher carriage of uninucleated cyst-producing Entamoeba species, compared to other Entamoeba species, identified at Twycross Zoo may be explained by the asymptomatic commensal carriage of a non-pathogenic Entamoeba species. These non-pathogenic species are less likely to be clinically identified; therefore, infected primates are less likely to receive amoebicidal treatment. Administration of amoebicidal drugs might have been the cause of the apparent increase in the prevalence of uninucleated cyst-producing Entamoeba species and Entamoeba coli in primates at Twycross Zoo. Re-establishment of gastrointestinal microflora, following treatment with amoebicidal agents, may have favoured the growth of the commensal populations of the octonucleated cyst-producing Entamoeba species (E. coli) in treated primates, as confirmed by microscopic examination of stool samples. In line with this assumption is the reported high frequency of the commensal uninucleated and octonucleated cyst-producing commensal Entamoeba species in primate populations (Tachibana et al., 2000; Petrášová et al., 2010). Alternatively, this may be explained by the development of metronidazole resistance in these uninucleated and octonucleated cyst-producing Entamoeba species.

The difference in the prevalence of Entamoeba among zoos (Table 5) can be explained by the differences in biosecurity and precautionary measures taken to prevent parasitic disease transmission. All zoos participated in the study already implement routine disinfection programmes (personal communication with Zoos). However, additional precautionary measures are needed in order to prevent the transmission of infection between enclosures including hygienic food preparation, provision of potable water, and disinfection of keeper footwear, over-clothing, hands, and cleaning equipment between enclosures. Effective drainage and water microfiltration within enclosures is also critical. Proactive pest control measures reduce arthropod vectors transporting infective cysts between enclosures (Pang, Chang & Chang, 1993; Denver, 2008). Additionally, avoiding mixed primate exhibits reduces the transmission of amoebiasis between NHP of different susceptibility. The same measure may help to prevent zoonotic transmission to zoo visitors. Unfortunately limited time and financial resources often result in deficiencies in one or more of these measures.

Methods for Entamoeba identification have been undergoing rapid change over the past decade and molecular phylogenetic techniques are rapidly becoming the procedures of choice (Levecke et al., 2010; Stensvold et al., 2011). PCR amplification of the 18S rDNA gene directly from a sample of mixed microbiota alleviates the need for culturing Entamoeba (Levecke et al., 2010), and once DNA is prepared, there are no biohazard dangers. rDNA-based molecular phylogenetic techniques were used to identify the Entmaoeba species detected in the faecal samples from NHP in the present study. Sequences from E. histolytica and E. dispar obtained in the study were identical to previously reported sequences in Genbank AB197936 and AB282661, respectively. Phylogenetic analysis of the partial-length 18SrDNA sequence showed that the uninucleate amplicons from Twycross Zoo were all E. polecki.

The uninucleated-cyst-producing Entamoeba infecting humans E. polecki species complex has been found to encompass four subtypes (ST1–ST4) (Stensvold et al., 2011). E. polecki ST1 (previously given to E. polecki in pigs); ST2 (E. chattoni from non-human primates); ST3 (E. struthionis from pigs and ostriches); and ST4 (restricted to humans; unlikely to be zoonotic); indicating low host specificity of ST1 and ST3. Comparison between sequences obtained in the present study and reference sequences obtained from GenBank for each of the four E. polecki subtypes indicated that sequences of E. polecki obtained in the present study from NHPs [Woolly Monkey (Lagothrix lagotricha), Eastern Javan Langur (Trachypithecus auratus auratus), Golden-headed Lion Tamarin (Leontopithecus chrysomelas), and Black Howler Monkey (Alouatta caraya)] formed a phylogenetic cluster (Fig. 2) with isolates of E. polecki subtype 4 reported in Homo sapiens from Africa, Asia and Europe (Stensvold et al., 2011). Given the reported high specificity of E. polecki subtype 4 to humans, the similarity between sequences obtained from NHPs from Twycross zoo in the present study with E. polecki ST4 sequences obtained from Homo sapiens suggest a zoonotic potential. However, more analysis is needed before any suggestion about the zoonotic implication of the isolates obtained in this study to be made. A group of uninucleate Entamoebas (referred to as Entamoeba RL3), phylogenetically distant from the E. polecki complex, have been reported from Francois Langur (Trachypithecus francoisi) from Twycross Zoo in England (Stensvold et al., 2011). Interestingly, sequences of this Entamoeba RL3 group did not seem to share similarity with the sequences of uninucleated-cyst-producing E. polecki obtained in the present study from NHPs from the same Zoo.

Sequence data (Table 4) also suggest that a common source asymptomatic infection with the uninucleated cyst-producing Entamoeba species, E. polecki, at Twycross Zoo may have propagated through many primate enclosures. This study did not examine the prevalence of E. nutalli, an emerging species currently seeming to be prevalent in NHPs (Tachibana et al., 2013). Since E. nuttalli has been associated with symptomatic carriage, and appears to be restricted in host distribution to NHPs, it would be interesting to know whether any of the animals sampled in the present study harboured E. nuttalli. Thus, further studies are needed to establish the prevalence of this important species in NHPs in the United Kingdom and its zoonotic risk to public health.

Conclusion

This is the first study to report the prevalence of Entamoeba infection in captive NHPs in the United Kingdom. Data collected from six zoos suggests a notable prevalence of Entamoeba infection in NHPs in UK. DNA sequencing of positive stool samples revealed three main species of Entamoeba, E. histolytica, E. dispar and E. polecki ST4 circulating in the zoo’s environment in the UK. Some Entamoeba species can have zoonotic potential, thus can constitute a risk for humans who are in close contact with primates.

Supplemental Information

Supplemental Information Raw Data

Click here for additional data file.

The following persons and facilities provided support and samples: Nic Masters, primate keepers, research collaborators and curators from all participating zoological parks. John Williams and Emma Victory from Diagnostic Parasitology Laboratory, London School of Hygiene and Tropical Medicine provided technical advice on PCR optimization experiments.

Additional Information and Declarations

Competing Interests

Author Contributions

Animal Ethics

DNA Deposition

Data Deposition

The authors declare there are no competing interests.

Carl S. Regan conceived and designed the experiments, performed the experiments, wrote the paper, prepared figures and/or tables, reviewed drafts of the paper.

Lisa Yon contributed reagents/materials/analysis tools, reviewed drafts of the paper.

Maqsud Hossain analyzed the data, contributed reagents/materials/analysis tools, reviewed drafts of the paper.

Hany M. Elsheikha conceived and designed the experiments, performed the experiments, analyzed the data, contributed reagents/materials/analysis tools, wrote the paper, prepared figures and/or tables, reviewed drafts of the paper.

The following information was supplied relating to ethical approvals (i.e., approving body and any reference numbers):

The study was approved by The University of Nottingham (UK) School of Veterinary Medicine and Science (SVMS) Ethical Review Committee.

The committee passed this study as not requiring any further ethical review.

The following information was supplied regarding the deposition of DNA sequences:

GenBank KJ149294, KJ149295, KJ149296.

The following information was supplied regarding the deposition of related data:

Nottingham ePrints Archive (http://eprints.nottingham.ac.uk/).

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
