# Peer review of "Prevalence of Entamoeba species in captive primates in zoological gardens in the UK"

_PeerJ, doi:10.7717/peerj.492_

## Round 0.1 · original submission · Minor Revisions

· Academic Editor

Minor Revisions

I would like to request a modification of organizational structure of Material & Methods and Results Sections aiming to turn them more attractive to the readers. Please, create subitens such as Study Areas and Sampling Design, Parasite Identification, Molecular Analyses, Phylogenetic and Statistical Analyses intead Experiment 1 and Experiment 2.

·

Basic reporting

No comments

Experimental design

No comments

Validity of the findings

No comments

Additional comments

The paper by Regan et al. is based on a good data set and clear. Reported findings are relevant. However, some changes should be made prior to publication.

1) Lines 76-77: should be deleted. These methods do exist and are nowadays commonly used;
2) Line 228: why “30 primates sampled”?? In the whole manuscript (also only few lines before at Line 225) the number 37 is reported;
3) Lines 228-229: I don’t think that Levecke et al recorded a 36% prevalence of E. histolytica!!! Probably they found a 36% prevalence of E. histolytica group. This is a very relevant difference, a matter of life or death I would say!! Please, write properly;
4) Lines 304-308: should be deleted. Too obvious and redundant. The concept is already clear at line 304;
5) Lines 316-321: This phrase doesn’t fit with the explanation of the higher prevalence of uninucleated cyst-producing Entamoeba species and E. coli; re-infection for transmission from other primates of the zoo could regard all Entamoeba species, not only those presenting higher prevalence post treatment. This phrase should be deleted;
6) Lines 388-392: should be deleted. Too long, too speculative, obvious and useless;
7) Table 1: Why scientific names only for some species? This is very strange. In general, as this is a paper for an international journal, scientific names are much more important than common ones (than could be totally omitted). Please, provide scientific names correctly.

Reviewer 2 ·

Basic reporting

Interesting research paper reporting notable prevalence of Entamoeba infection in captive non-human primates in the United Kingdom. Since some Entamoeba species have zoonotic potential, the observations are of importance with respect to public health.

Experimental design

The submission describes original primary research. There are only some minor points that should be improved:
1. Abstract is too long and should be significantly shortened.
2. The manuscript should be better organized (avoid dividing the paper into the parts titled Experiment 1 and 2).

Validity of the findings

The data are well presented and adequately discussed.

---

## Round 0.2 · accepted · Accept

· Academic Editor

Accept

The manuscript was adequately reorganized and all the reviewer`s comments were well addressed.